# Accelerating Certified Robustness Training via Knowledge Transfer

**Pratik Vaishnavi**
Stony Brook University
`pvaishnavi@cs.stonybrook.edu`

**Kevin Eykholt**
IBM Research
`kheykholt@ibm.com`

**Amir Rahmati**
Stony Brook University
`amir@cs.stonybrook.edu`

## Abstract

Training deep neural network classifiers that are *certifiably* robust against adversarial attacks is critical to ensuring the security and reliability of AI-controlled systems. Although numerous state-of-the-art certified training methods have been developed, they are computationally expensive and scale poorly with respect to both dataset and network complexity. Widespread usage of certified training is further hindered by the fact that periodic retraining is necessary to incorporate new data and network improvements. In this paper, we propose Certified Robustness Transfer (CRT), a general-purpose framework for reducing the computational overhead of any certifiably robust training method through knowledge transfer. Given a robust teacher, our framework uses a novel training loss to transfer the teacher's robustness to the student. We provide theoretical and empirical validation of CRT. Our experiments on CIFAR-10 show that CRT speeds up certified robustness training by $8\times$ on average across three different architecture generations while achieving comparable robustness to state-of-the-art methods. We also show that CRT can scale to large-scale datasets like ImageNet.

## 1 Introduction

Deep Neural Networks (DNNs) are susceptible to adversarial evasion attacks [31, 9], that add a small amount of carefully crafted imperceptible noise to an input to reliably trigger misclassification. As a defense, numerous training methods have been proposed [25, 40, 35] to grant empirical robustness to a DNN. But in the absence of any provable guarantees for this robustness, these defenses were frequently broken [1, 32]. These failures have motivated the development of training methods that grant certifiable/provable robustness to a classifier, hence safeguarding it against all attacks (known or unknown) within a pre-determined threat model. Such methods are broadly categorized as either *deterministic* or *probabilistic* [23]. Deterministic robustness training methods [12, 26, 33, 34, 28, 10, 41, 30] rely on computing provable bounds on the output neurons of a classifier for a given perturbation budget in the input space. However, the deterministic robustness guarantees provided by these methods come at a high computational cost. Probabilistic robustness training methods address this limitation by providing highly probable (*e.g.,* with 0.99 probability) robustness guarantees at a greatly reduced computational cost. Within this category, *randomized smoothing*-based methods [19, 3, 29, 22, 20, 7, 37, 39, 16, 15] are considered the state-of-the-art for certifiable robustness in the $\ell_2$-space. Even so, these training methods remain an order of magnitude slower than standard training. In commercial applications where constant model re-deployment occurs to provide improvements (see Figure 1), re-training using computationally expensive methods is burdensome.

36th Conference on Neural Information Processing Systems (NeurIPS 2022).

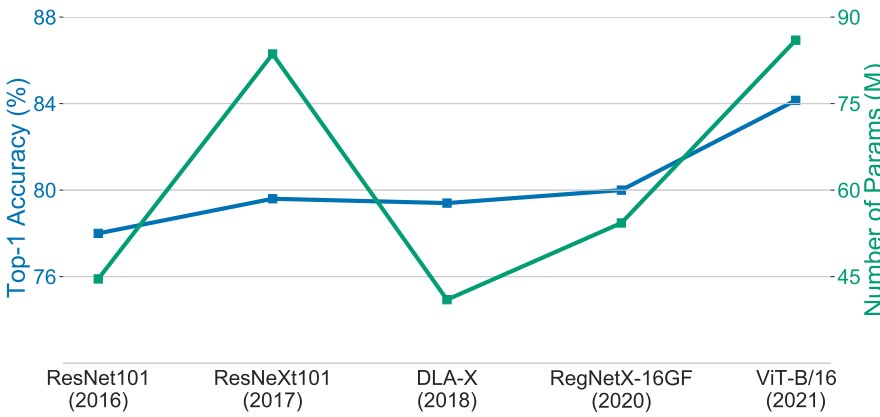

Figure 1: Evolution of DNN architectures on the ImageNet dataset. We plot the performance (top-1 accuracy) and the number of parameters of a few popular architectures (year of release is noted in brackets). Newer generations attempt to improve performance and/or reduce network parameters.

In this work, we reduce the training overhead of randomized smoothing-based robustness training methods with minimal impact on the robustness achieved. We propose Certified Robustness Transfer (CRT), a knowledge transfer framework that significantly speeds up the process of training $\ell_2$ certifiably robust image classifiers. Given a pre-trained classifier that is certifiably robust (*i.e., teacher*), CRT trains a new classifier (*i.e., student*) that has comparable levels of robustness in a fraction of the time required by state-of-the-art methods. CRT brings down the cost of training certifiably robust image classifiers to be comparable to standard training while preserving state-of-the-art robustness. On CIFAR-10, CRT speeds up training by an average of $\mathbf{8}\times$ across three different architecture generations compared to a state-of-the-art robustness training method [15]. Furthermore, we show that state-of-the-art robustness training is only necessary to train the initial classifier. Afterward, CRT can be continuously reused to transfer robustness in order to expedite future model re-deployments and greatly reduce costs associated with computational resources. Our **contributions** can be summarized as follows:

- We present Certified Robustness Transfer (CRT), the first framework, to our knowledge, that can transfer the robustness of a certifiably robust teacher classifier to a new student classifier. CRT greatly reduces the time required to train certifiably robust image classifiers relative to existing state-of-the-art methods while achieving comparable or better robustness.

- We provide a theoretical understanding of CRT, showing how our approach of matching outputs enables robustness transfer between the student and teacher irrespective of the certified robustness training method used to train the teacher.

- On CIFAR-10, we show that CRT trains certifiably robust classifiers on average $8\times$ faster than a state-of-the-art method while having comparable or better Average Certified Radius (by $8\%$ in the best case). Furthermore, CRT reduces the cumulative computational cost of training three classifiers by $87.84\%$.

- We also show that CRT can be reused in a recursive manner, thus supporting a continuous re-deployment scenario (*e.g.,* in commercial applications). Finally, we show that CRT remains effective on a large-scale dataset, ImageNet.

## 2 Background

In this section, we briefly introduce certified robustness and discuss notable existing methods for training certifiably robust image classifiers using randomized smoothing.

### 2.1 Preliminaries

***Problem Setup.*** Consider a neural network classifier $f$ parameterized by $\theta$ (denoted $f_\theta$) trained to map a given input $x \in \mathbb{R}^d$ to a set of discrete labels $\mathcal{Y}$ using a set of *i.i.d.* samples $\mathcal{S} = \{(x_1, y_1), (x_1, y_1), \cdots, (x_n, y_n)\}$ drawn from a data distribution $\mathcal{D}$. The output of the classifier can be written as $f_\theta(x) = \arg\max_{c \in \mathcal{Y}} z_\theta^c(x)$. Here $z_\theta(x)$ is the softmax output of the classifier and $z_\theta^c(x)$ denotes the probability that image $x$ belongs to class $c$.

***Certified Robustness via Randomized Smoothing.*** The robustness of the classifier $f_\theta$ for a given input pair $(x, y)$ is defined using the radius of the largest $\ell_2$ ball centered at $x$ within which $f_\theta$ has a constant output $y$. This radius is referred to as *robust radius* and it can mathematically be expressed as:

$$R(f_\theta; x, y) = \begin{cases} \inf_{f_\theta(x') \neq f_\theta(x)} \|x' - x\|_2 & \text{, when } f_\theta(x) = y \\ 0 & \text{, when } f_\theta(x) \neq y \end{cases} \tag{1}$$

Within this $\ell_2$-neighborhood of $x$, $f_\theta$ is considered to be *certifiably robust*. Therefore, to improve the robustness of a classifier, one needs to maximize this robust radius corresponding to any point sampled from the given data distribution. Directly maximizing the robust radius of a DNN classifier is an NP-hard problem [17]. Therefore, several prior works attempt to derive a lower bound for the robust radius [21, 19, 3]. This lower bound, often termed as the *certified radius*, satisfies the following condition: $0 \leq CR(f_\theta; x, y) \leq R(f_\theta; x, y)$, for any $f_\theta$, $(x, y)$. In this paper, we utilize the certified robustness framework derived by Cohen *et al.* [3] using randomized smoothing. Given a classifier $f_\theta$, they first define the smooth classifier $g_\theta$ as:

**Definition 2.1.** For a given (base) classifier $f_\theta$ and $\sigma > 0$, the smooth classifier $g_\theta$ corresponding to $f_\theta$ is defined as follows:

$$g_\theta(x) = \arg\max_{c \in \mathcal{Y}} P_{\eta \sim \mathcal{N}(0, \sigma^2 I)}(f_\theta(x + \eta) = c) \tag{2}$$

Simply put, $g_\theta$ returns the class $c$, which has the highest probability mass under the Gaussian distribution $\mathcal{N}(x, \sigma^2 I)$. Using Theorem 2.2, they proved that if the smooth classifier correctly classifies a given input $x$, it is certifiably robust at $x$. They also provided an analytical form of the $\ell_2$ certified radius at $x$.

**Theorem 2.2.** *Let $f_\theta : \mathbb{R}^d \mapsto \mathcal{Y}$ be a classifier and $g_\theta$ be its smoothed version (as defined in Definition 2.1). For a given input $x \in \mathbb{R}^d$ and corresponding ground truth $y \in \mathcal{Y}$, if $g_\theta$ correctly classifies $x$ as $y$,* i.e.,

$$P_\eta(f_\theta(x + \eta) = y) \geq \max_{y' \neq y} P_\eta(f_\theta(x + \eta) = y') \tag{3}$$

*then $g_\theta$ is provably robust at $x$ within the certified radius $R$ given by:*

$$CR(g_\theta; x, y) = \frac{\sigma}{2}[\Phi^{-1}(P_\eta(f_\theta(x + \eta) = y)) - \Phi^{-1}(\max_{y' \neq y} P_\eta(f_\theta(x + \eta) = y'))] \tag{4}$$

*where $\Phi$ is the c.d.f. of the standard Gaussian distribution.*

This certified radius is a *tight* lower bound of the robust radius defined in Equation 1, *i.e.,* it is impossible to certify $g_\theta$ at $x$ for a radius larger than $CR$.

## 2.2 Training Methods for Maximizing Certified Radius

In addition to the theoretical framework discussed above, Cohen *et al.* [3] also propose a simple yet effective method for training the base classifier in a way that maximizes the $\ell_2$ certified radius of the smooth classifier, as expressed in Equation 4. We include an evaluation of their method in Appendix **??**. Following their work, several other works build upon the randomized smoothing framework and propose training methods that better maximize the $\ell_2$ certified radius of the smooth classifier. Salman *et al.* [29] proposed combining adversarial training [25] with randomized smoothing (called SmoothAdv). They adapted the vanilla PGD attack to target the smooth classifier $g_\theta$ instead of the base classifier $f_\theta$. Zhai *et al.* [39] proposed a new robustness loss, a hinge loss that enforces maximization of the soft approximation of the certified radius. Their method (called MACER) is faster than SmoothAdv as it does not use adversarial training. More recently, Jeong *et al.* [15] proposed training with a convex combination of samples along the direction of adversarial perturbation for each input to regularize over-confident predictions. Their method (called SmoothMix) is the current state-of-the-art in the domain of $\ell_2$ certified robust image classifiers. Finally, we note the Consistency regularization method proposed by Jeong *et al.* [16], which adds a regularization loss to existing methods that helps better maximize the certified radius.

Table 1: Training on CIFAR-10 using a ResNet110 classifier on a single Nvidia V100 GPU. State-of-the-art robustness training methods significantly slow down training compared to standard training.

| METHOD | TRAINING SLOWDOWN FACTOR |
|---|---|
| SMOOTHADV | $46.20\times$ |
| MACER | $20.86\times$ |
| SMOOTHMIX | $4.97\times$ |

## 3 Maximizing Certified Radius via Knowledge Transfer

Although prior works have proposed methods for increasing the certified radius of the smooth classifier, their training overhead is significant, making them much slower than standard training. As we show in Table 1, training a certifiably robust ResNet110 classifier to convergence using SmoothAdv, MACER, and SmoothMix is $46.20\times, 20.86\times$, and $4.97\times$ slower, respectively, compared to training a non-robust classifier with standard training.

Given constant innovations in architecture design (Figure 1) and the influx of new data, which may result in various tweaks to deployed networks that elicit retraining, the large overhead of state-of-the-art robustness training methods makes preserving certified robustness across model re-deployment difficult. Therefore, we propose Certified Robustness Transfer (CRT), a training method that improves the usability of certified robustness training methods by dramatically reducing their training overhead while preserving the certified robustness. Given the *base classifier* of a pre-trained certifiably robust *smooth classifier*, we leverage the knowledge transfer framework to guide the training of a new base classifier (and its associated robust smooth classifier).[1] In this section, we describe our method and provide theoretical justification for its effectiveness.

### 3.1 Transferring Certified Robustness

From Equation 4, it follows that training the base classifier to maximize $P_\eta(f_\theta(x + \eta) = y)$ for any given input $x$ will result in the maximization of the certified radius associated with the smooth classifier, provided Equation 3 is satisfied. Thus, for the base classifier $f_\theta(x)$, our goal is to maximize the following quantity over the training set:

$$\sum_{i=1}^{n} \mathbb{E}_\eta \mathbf{1}[f_\theta(x_i + \eta) = y_i] \approx \sum_{i=1}^{n} \mathbb{E}_\eta[z_\theta^{y_i}(x_i + \eta)] \tag{5}$$

In the above equation, like prior works [3, 29, 39], we leverage the fact that the softmax output of a classifier can be treated as a continuous and differentiable approximation of its $\arg\max$ output. Methods like SmoothAdv [29], MACER [39] and SmoothMix [15] that target $\ell_2$ certifiable robustness propose training objectives that maximize this term.

Now, suppose we have a pre-trained base classifier $f_\phi$. It follows that $\mathbb{E}_\eta[z_\phi^y(x + \eta)] \geq 0$. Through straightforward algebraic manipulations (see Appendix **??**), we derive the following lower bound:

$$\sum_{i=1}^{n} \mathbb{E}_\eta[z_\theta^{y_i}(x_i + \eta)] \geq -\sum_{i=1}^{n} \mathbb{E}_\eta[z_\phi^{y_i}(x_i + \eta) - z_\theta^{y_i}(x_i + \eta)] \tag{6}$$

That is to say that, for a given input $x_i$, if we minimize the difference between the softmax outputs of the teacher and the student ($f_\phi$ and $f_\theta$) corresponding to the correct label $y_i$, we will maximize Equation 5 for the student. However, to ensure that the student has a non-trivial certified radius, we must also ensure that Equation 3 is satisfied. If we assume that Equation 3 holds true for the teacher (*i.e.,* the base classifier of a certifiably robust smooth classifier), this condition can also be achieved for the student by matching the overall softmax output of the student to that of the teacher.

### 3.2 Certified Robustness Transfer (CRT)

Based on the previous discussion, we now describe our method for training a certifiably robust classifier through knowledge transfer. First, we obtain a pre-trained base classifier $f_\phi$, which has been trained using a randomized smoothing based robustness training method as this maximizes

---

[1]If no pre-trained classifier is available, we first train an architecture of lower complexity (*i.e.,* fast to train) compared to the target architecture (Section 5.1).

---
**Algorithm 1** Certified Robustness Transfer (CRT)
___
1: **Input:** Training data distribution $\mathcal{D}$, certifiably robust teacher base classifier $f_\phi$, noise level $\sigma$, total training iterations $\mathcal{T}$, learning rate $\alpha$
2: **Output:** Certifiably robust student base classifier $f_\theta$
3: $\theta \leftarrow$ random initialization
4: $i \leftarrow 0$
5: **while** $i < \mathcal{T}$ **do**
6:     From $\mathcal{D}$, sample a batch of inputs $\{x_1, x_2, \cdots, x_n\}$.
7:     From $\mathcal{N}(0, \sigma^2 I)$, generate a batch of Gaussian noise samples $\{\eta_1, \eta_2, \cdots, \eta_n\}$.
8:     $l_i \leftarrow \frac{1}{n} \sum_{j=1}^{n} \|z_\phi(x_j + \eta_j) - z_\theta(x_j + \eta_j)\|_2$
9:     $\theta \leftarrow \theta - \alpha \cdot \nabla_\theta l_i$
10:    $i \leftarrow i + 1$
11: **end while**
___

$\mathbb{E}_\eta[z_\phi^y(x + \eta)]$. Next, we use $f_\phi$ as a teacher to train a new student base classifier $f_\theta$. The student is trained to match the output of the teacher. In doing so, we can maximize the certified radius of the associated smooth classifier $g_\theta$ (Equation 6), as well as ensure that Equation 3 is satisfied. We describe our implementation in Algorithm 1. Given a batch of inputs, we first perturb them with additive Gaussian noise. Next, we compute the $\ell_2$ distance between the student and the teacher's outputs for these Gaussian perturbed inputs. This distance serves as our loss function, and we update the parameters of the student to minimize this loss. At test time, the classifier $f_\theta$ is converted to its smooth version $g_\theta$ following Definition 2.1.

### 3.3 Prior Works on Robustness Transfer

Several prior works have examined transferring adversarial robustness between classifiers, but these works have been limited to transferring empirical rather than certified robustness [2, 8, 14, 42, 43]. Of note is the work by Goldblum *et al.* [8] in which they combine adversarial training [25] with knowledge distillation [13]. They show that distilling knowledge from a large network to a small network improves its empirical robustness as compared to training the small network on its own, but their method makes no effort to improve the computational cost of adversarial training.

## 4 Evaluation

Our goal is to improve the usability of randomized smoothing based robustness training methods. In this section, we demonstrate how CRT enables the reuse of an existing certifiably robust classifier to train new certifiably robust classifiers at significantly reduced training cost compared to prior methods. In our first experiment, we train a ResNet110 classifier with a state-of-the-art method, *i.e.,* SmoothMix [15]), and use CRT to transfer its robustness to train several newer generation classifiers. In a second experiment, we recursively use CRT to train a newer generation classifier using the previous generation classifier that was also trained using CRT. In each experiment, we compare the certified robustness of classifier trained using CRT against a classifier trained using SmoothMix (Section 4.1). We find that classifiers trained using CRT are similarly robust as when trained using SmoothMix but only require a fraction of training time (Section 4.2). Our main results are generated using the CIFAR-10 dataset [18], but we also demonstrate the effectiveness of CRT on ImageNet [5] (Section 5.3). Both these datasets are open-source and free for non-commercial use.

*Architectures.* We use several popular DNN architectures that were proposed to either improve upon the visual recognition performance of the previous generation architectures or preserve performance while requiring significantly fewer parameters (or both). For the CIFAR-10 experiments, we use ResNet110 [11], ResNeXt29-2x64d [36], DLA [38], and RegNetX_200MF [27].[2]

*Training details.* All SmoothMix classifiers were trained using the code made available by the authors[3] and the hyperparameters reported by them [15]. All CRT classifiers were trained using Stochastic Gradient Descent till convergence (200 epochs), with a batch size of 128. Further hyperparameter details are available in Appendix **??**. Unless specified, we report results for noise

---
[2]We use the CIFAR-10 version of these architectures, code [MIT License]: `https://github.com/kuangliu/pytorch-cifar`.
[3]SmoothMix code [MIT License]: `https://github.com/jh-jeong/smoothmix`

level $\sigma = 0.25$ in the main paper. Additional results for higher noise levels $\sigma = 0.5$ and $1.0$ are reported in Appendix **??**.

***Evaluation Metrics.*** We report our results using two metrics. First, as done in prior work, we measure the **certified robustness** of a classifier based on (1) the *certified test accuracy* at $\ell_2$ radius $r$ [4], which is defined as the fraction of test set inputs that the smooth classifier classifies correctly within an $\ell_2$ ball of radius $r$ centered at each input, and (2) *average certified radius* (ACR), which is the average of the certified radius across all inputs in the test set:

$$ACR(g_\theta) = \frac{1}{n_{test}} \sum_{i=1}^{n_{test}} CR(g_\theta; x_i, y_i)$$

On CIFAR-10, we compute these metrics using the entire test set. Second, we measure **training time** of a classifier based on the *per-epoch time* and *total training time*. The total training time is computed once the model's loss has converged. All classifiers were trained on the same machine with a single Nvidia Titan V GPU.

### 4.1   Certified Robustness Comparison

***Standard CRT Training.*** Given a ResNet110 classifier trained using SmoothMix, we transfer its robustness to several newer generation classifiers. We compare the certified robustness of these classifiers with their SmoothMix trained versions. The results are summarized in Table 2. We observe that using CRT does not reduce the certified robustness of the trained classifier compared to training with SmoothMix. In fact, interestingly, CRT trained classifiers exhibit higher certified robustness compared to their SmoothMix baseline. Not only do CRT trained classifiers have higher ACR (improvement of $8.1\%$ in the best case), they also exhibit higher certified accuracy at different $\ell_2$ radii. Furthermore, CRT remains effective even as the generation gap between the student and the teacher increases. This implies that the same teacher can potentially be reused indefinitely, amortizing the teacher's training cost to a constant. These results empirically validate our theoretical justification of CRT. Finally, we note that in Table 2, the accuracy on clean inputs and ACR of CRT trained classifiers follow the same trend as in Figure 1, thus motivating the need for periodic model re-deployment to incorporate architectural improvements.

Table 2: The certified robustness of classifiers with different architectures trained on CIFAR-10 using SmoothMix [15] and CRT. We use CRT to transfer the robustness of a ResNet110 trained using SmoothMix. We report certified test accuracy at different values of $\ell_2$ radius and the Average Certified Radius (ACR). The architectures are sorted chronologically based on published date. The noise level $\sigma$ is set to $0.25$.

| ARCHITECTURE | 0.00 | 0.25 | 0.50 | 0.75 | ACR |
|---|---|---|---|---|---|
| SMOOTHMIX [15] | | | | | |
| RESNET110 [11] | 76.89 | 68.25 | 57.42 | 46.26 | 0.550 |
| RESNEXT29-2X64D [36] | 75.98 | 65.40 | 53.78 | 41.03 | 0.516 |
| DLA [38] | 77.72 | 68.53 | 57.69 | 45.56 | 0.551 |
| REGNETX_200MF [27] | 76.48 | 66.79 | 56.36 | 44.47 | 0.538 |
| CRT (RESNET110 TEACHER) | | | | | |
| RESNEXT29-2X64D [36] | 77.57 | 69.00 | 58.31 | 47.16 | 0.558 |
| DLA [38] | 77.31 | 68.91 | 58.26 | 46.34 | 0.554 |
| REGNETX_200MF [27] | 77.89 | 69.57 | 59.36 | 47.28 | 0.564 |

***Recursive CRT Training.*** We now explore the effectiveness of CRT if it is used recursively, *i.e.,* the newest generation is trained using a CRT trained classifier from the previous generation as the teacher. We begin with a ResNet110 trained using SmoothMix. Then, all subsequent classifiers are trained using CRT recursively and report the results in Table 3. The *chain length* measures the number of times CRT was used. For example, the DLA network, with a chain length of 2, is the result of using CRT

---

[4]Note that the certified accuracy at $r = 0$ represents the clean accuracy of the smooth classifier.

Table 3: The certified robustness of classifiers with different architectures trained on CIFAR-10 using CRT recursively. We report certified test accuracy at different values of $\ell_2$ radius and the Average Certified Radius (ACR). Here, the previous generation classifier is used to train the current generation one. Chain length represents the number times CRT was used in training. The noise level $\sigma$ is set to 0.25. CRT remains effective despite recursive use.

| ARCHITECTURE | CHAIN LENGTH | 0.00 | 0.25 | 0.50 | 0.75 | ACR |
|---|---|---|---|---|---|---|
| RESNEXT29-2X64D [36] | 1 | 77.57 | 69.00 | 58.31 | 47.16 | 0.558 |
| DLA [38] | 2 | 78.46 | 70.05 | 60.01 | 48.30 | 0.570 |
| REGNETX_200MF [27] | 3 | 78.16 | 69.00 | 58.69 | 47.00 | 0.559 |

twice: once to transfer the SmoothMix trained ResNet110 network's performance to the ResNeXt29-2x64d network and once to transfer the CRT trained ResNeXt29-2x64d network's performance to the DLA network. We observe that the certified robustness of the resulting classifiers remains high even with recursive use of CRT. The empirical results are to be expected given our theoretical understanding of CRT: In order to train a robust student, we only require that the teacher is already robust (*i.e.,* satisfies the condition of Theorem 2.2) *irrespective of the training method used to achieve robustness*. Thus, we expect CRT to remain effective even at longer chain lengths. In Section 5.2, we will highlight the relationship between the teacher's training method and the robustness of a CRT trained student.

## 4.2 Training Time Comparison

Having established that CRT effectively transfers certified robustness between classifiers, we now evaluate its training overhead. For comparison, we also evaluate the training overhead of SmoothMix. In Table 4, we report the per-epoch time and total time of training different architectures with each method. For brevity, we only compare the training time for standard CRT.[5] We observe that the per-epoch time of CRT is significantly lower than SmoothMix. Similarly, when trained until convergence, the total training time of CRT is significantly lower. Across the three architectures that we run our experiments on, CRT achieves an average epoch time speedup of $10.75\times$. Comparing overall training times, CRT speeds up training by, on average, $8.06\times$. If we consider the real-world scenario where the model has to be periodically redeployed to incorporate architectural improvements, the cumulative training time using SmoothMix is 96.21 hours as each new architecture is trained from scratch. With CRT, the cumulative time is reduced to 11.70 hours representing a $87.84\%$ savings in costs associated with computational resources.

***Teacher's availability.*** So far, we assumed the availability of a certifiably robust teacher (ResNet110). We argue that this is a reasonable assumption as the amortized cost associated with the one-time training of a robust teacher is negligible across many generations of the model. Regardless, in Section 5.1, we examine a scenario where the teacher is unavailable. Under this scenario, we demonstrate how CRT can be used to speedup the training of ResNet110 for use as teacher.

Table 4: Training time statistics for SmoothMix and CRT. We report the mean and 95% confidence interval computed over all training epochs. CRT is on average $8\times$ faster than SmoothMix across all three architectures.

| ARCHITECTURE | SMOOTHMIX [15] | | CRT (RESNET110 TEACHER) | |
|---|---|---|---|---|
| | EPOCH TIME (S) | TOTAL TIME (H) | EPOCH TIME (S) | TOTAL TIME (H) |
| RESNET110 [11] | $455.55 \pm 1.17$ | 18.98 | - | - |
| RESNEXT29-2X64D [36] | $1085.09 \pm 0.50$ | 45.21 | $86.41 \pm 0.11$ | 4.80 |
| DLA [38] | $854.41 \pm 0.09$ | 35.60 | $62.24 \pm 0.40$ | 3.46 |
| REGNETX_200MF [27] | $369.42 \pm 0.51$ | 15.39 | $61.92 \pm 0.30$ | 3.44 |

---

[5]Recursive CRT differs in time by an insignificant factor due to forward pass through a different teacher.

# 5 Discussion

In this section, we address the standout concerns about CRT. The section layout is as follows: in Section 5.1, we discuss the scenario in which a certifiably robust teacher is not readily available and demonstrate how CRT can still speed up robustness training; in Section 5.2, we examine how the method used to train the teacher affects the robustness of the student; in Section 5.3, we study the scalability of CRT using the ImageNet dataset; in Section 5.4, we compare CRT with a closely related prior work on fast certified robustness training, *i.e.,* Consistency regularization [16]; in Section 5.5, we discuss the limitations of CRT; in Section 5.6, we address the broader impact of CRT.

## 5.1 Teacher Not Available

We've designed CRT under the assumption that a certifiably robust teacher is already available. However, even if a certifiably robust teacher is not available, CRT can still speed up training. Given a certifiable robust training method and a large network architecture, we can reduce the training overhead by robustly training a comparatively smaller network first. Then, we can use CRT to transfer the robustness of the small network to a larger network. In Table 5, we present results for such a process. First, we trained a ResNet20 network using SmoothMix, then we used CRT to train a ResNet110 network. We compare the robustness of a ResNet110 trained using this process with one trained using SmoothMix. As we can see, the CRT ResNet110 network has comparable robustness with the SmoothMix ResNet110 network. However, even when adding the teacher and student training times, CRT still speeds up training by approximately $2\times$ relative to SmoothMix.

Table 5: Certified robustness and total time of a ResNet110 classifier trained on CIFAR-10 using SmoothMix and CRT. For CRT, we train a ResNet20 teacher first using SmoothMix and report total time as the time taken to train the teacher and the student. The noise level $\sigma$ is set to $0.25$. The ResNet110 trained using CRT achieves an ACR comparable to the SmoothMix ResNet110 while achieving a $\sim 2\times$ speedup in total training time.

| METHOD | 0.00 | 0.25 | 0.50 | 0.75 | ACR | TOTAL TIME (H) |
|--------|------|------|------|------|-----|----------------|
| SMOOTHMIX [15] | 76.89 | 68.25 | 57.42 | 46.26 | 0.550 | 18.98 |
| CRT (RESNET20 TEACHER) | 75.68 | 67.20 | 56.30 | 44.83 | 0.540 | 10.07 |

## 5.2 Teacher Training Method

We train a ResNet20 classifier using MACER [39], SmoothAdv [29], and SmoothMix [15]. For MACER and SmoothAdv training, we use the code made available by the authors[6,7] and the hyperparameters reported by them. Using CRT, we transfer the robustness of each teacher to a ResNet110 classifier. The results are reported in Table 6. For reference, we also report robustness of a ResNet110 network trained independently using the chosen robustness training methods. Overall, we observe a slight variation in the robustness of the CRT trained networks depending on the teachers training method. Based on Equation 6, this is expected as maximizing the teacher's performance will in turn maximize the performance of the student. Our empirical results align with this reasoning: the MACER teacher was the least robust of the three methods, and its student is similarly the least robust of the students. However, in all cases, the CRT trained network obtained certified robustness comparable to its teacher.

## 5.3 Scalability

Here, we study the effectiveness of CRT on a large-scale dataset, *i.e.,* ImageNet. For this purpose, we train ResNet18 classifiers using three certified robustness training methods (MACER, SmoothAdv, and SmoothMix). Next, we transfer their robustness to a ResNet50 classifier. The results were generated on a 500 sample test set (following prior works [29, 39, 15]) and are summarized in Table 7. For reference, we also report robustness of a ResNet50 network trained independently using the chosen robustness training methods. In all cases, we observe that students achieve certified robustness comparable to their respective teachers. Therefore, CRT remains effective even on a more complex dataset.

---

[6]MACER code [No license available]: `https://github.com/RuntianZ/macer`
[7]SmoothAdv code [MIT License]: `https://github.com/Hadisalman/smoothing-adversarial`

Table 6: For CIFAR-10 dataset, certified robustness achieved on training the CRT teacher (ResNet20) with different methods. The student classifier is ResNet110. For reference, we also report robustness of ResNet110 trained independently using chosen methods. The noise level $\sigma$ is set to 0.25. Students attain comparable ACR to their respective teachers.

| TEACHER (RESNET20) | | | STUDENT (RESNET110) | |
|---|---|---|---|---|
| **TRAINING METHOD** | **ACR** | | **TRAINING METHOD** | **ACR** |
| SMOOTHADV [29] | 0.531 | | CRT | 0.519 |
| MACER [39] | 0.507 | $\rightarrow$ | CRT | 0.528 |
| SMOOTHMIX [15] | 0.522 | | CRT | 0.540 |
| | | | SMOOTHADV [29] | 0.547 |
| STUDENT TRAINED DIRECTLY | | | MACER [39] | 0.531 |
| | | | SMOOTHMIX [15] | 0.550 |

Table 7: ImageNet results using CRT and three robustness training methods. We report both the ACR of the ResNet18 teacher and its ResNet50 student. For reference, we also report robustness of ResNet50 trained independently using chosen methods. The noise level $\sigma$ is set to 0.5. Students attain comparable ACR to their respective teachers.

| TEACHER (RESNET18) | | | STUDENT (RESNET50) | |
|---|---|---|---|---|
| **TRAINING METHOD** | **ACR** | | **TRAINING METHOD** | **ACR** |
| SMOOTHADV [29] | 0.684 | | CRT | 0.684 |
| MACER [39] | 0.574 | $\rightarrow$ | CRT | 0.576 |
| SMOOTHMIX [15] | 0.653 | | CRT | 0.661 |
| | | | SMOOTHADV [29] | 0.820 |
| STUDENT TRAINED DIRECTLY | | | MACER [39] | 0.653 |
| | | | SMOOTHMIX [15] | 0.799 |

## 5.4 Comparison with Consistency Regularization [16]

In Section 4, we compared CRT against SmoothMix as it has state-of-the-art ACR. However, another closely related work was recently published by Jeong & Shin [16], which shows state-of-the-art ACR and potential training time improvements. They proposed a consistency regularization loss that improves the certified robustness of smooth classifiers by enforcing the base classifier's soft outputs to be consistent across multiple noisy copies of a given input. Therefore, their additional computational overhead scales linearly with the number of noisy samples used to compute the consistency loss. With respect to computational overhead, CRT adds only one forward pass, *i.e.,* the pass through the teacher. When paired with Gaussian data augmentation training, their regularization loss significantly improves the certified robustness of a smooth classifier. By applying their regularization loss over only two noisy copies of the input, they can achieve better certified robustness than prior state-of-the-art robustness training methods like MACER [39] and SmoothAdv [29].

The key difference between CRT and consistency regularization is in the training overhead when combined with other state-of-the-art certified training methods. Consistency regularization augments classifier training with an additional loss term. Therefore, their training overhead is dominated by the training method selected. In their experiments, they focused on Gaussian data augmentation, which adds little to no training overhead relative to standard training. However, if a more computationally intensive method was selected (*e.g.,* MACER), they remark their training overhead would dramatically increase. With respect to CRT, if a teacher is available (*i.e.,* a previous generation model), the overhead of CRT is agnostic to the training method. If it is not available, we demonstrated in Section 5.1, that CRT can still greatly reduce training overhead. For interested readers, we include results for transferring robustness from a teacher trained using Consistency regularization in Appendix **??**.

### 5.5 Limitations

In this paper, we use probabilistic certified robustness methods as they rely on Theorem 2.2 and, thus, are designed to maximize the certified radius (Equation 4). We found that deterministic methods (*e.g.,* CROWN-IBP [41]) impose a stricter training requirement on the base teacher classifier. For a given input, deterministic training methods require the base classifier to be correct for all inputs within the $\ell_2$-norm ball, rather than simply be likely to correctly classify inputs within the $\ell_2$-norm ball. This restriction lowers the potential ACR of the smooth teacher classifier, which also lowers the ACR of the student trained using CRT. For example, when using CROWN-IBP [41] to train a ResNeXt base classifier, the ACR for the corresponding smooth classifier is only 0.064. When transferring the robustness of this ResNeXt classifier to a WideResNet34-10 student, we get an ACR of 0.065.

Additionally, we note that the classifier architectures we present in the paper are restricted to CNNs. Recently, a new class of transformer-based image classifiers [6, 24, 4] have been proposed that show improved performance over CNN classifiers. We briefly studied the effectiveness of CRT when transferring robustness between CNN and transformer architectures using ViT [6] and present the results in Appendix **??**, but further exploration is needed. Finally, CRT has only been studied using the $\ell_2$ norm and image data due to the limitations of current certified robustness training methods.

### 5.6 Broader Impacts

As we have shown, our work improves the efficiency of training certifiably robust classifiers, in an effort to improve the security of AI-powered systems. Beyond the broad negative societal impacts of machine learning, we are not aware of any impacts specific to our work.

## 6 Conclusion

In this paper, we proposed the first general-purpose framework to speed up the training of certifiably robust classifiers using knowledge transfer and randomized smoothing. Our proposed method, Certified Robustness Transfer (CRT) enables transferring the certified robustness of a classifier to another classifier at a cost comparable to standard training. We provided a theoretical understanding of CRT and provided empirical evidence of its effectiveness. On CIFAR-10, we showed that across several generations of classifier architectures, CRT trained classifiers $8\times$ faster than when using a state-of-the-art training method, while achieving comparable or better certified robustness. Furthermore, CRT can reduce the training overhead of certified robustness training methods even when an initial robust classifier is not present. The use of machine learning in security and safety critical environments motivates a need for models with certifiably robust performance, but the training overhead of existing certified robustness training methods inhibits their usability. Our work addresses this issue, especially for commercial applications where periodical model re-deployment is inevitable.

## Acknowledgement

This work was supported by the Office of Naval Research under grants N00014-20-1-2858 and N00014-22-1-2001, Air Force Research Lab under grant FA9550-22-1-0029, and NVIDIA 2018 GPU Grant. Any opinions, findings, or conclusions expressed in this material are those of the authors and do not necessarily reflect the views of the sponsors.

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
