# A    Justification for CRT Training Objective

Following from the discussion in Section 3.1, we want to maximize $\mathbb{E}_\eta[z_\theta^y(x + \eta)]$. Given a non-negative random variable $t = \mathbb{E}_\eta[z_\phi^y(x + \eta)]$, we have

$$\mathbb{E}_\eta[z_\theta^y(x + \eta)] \geq \mathbb{E}_\eta[z_\theta^y(x + \eta)] - t$$

Here $z_\phi$ is the soft output associated with classifier $f_\phi$ that has been trained independently of $f_\theta$. Therefore, we have

$$\mathbb{E}_\eta[z_\theta^y(x + \eta)] \geq \mathbb{E}_\eta[z_\theta^y(x + \eta)] - \mathbb{E}_\eta[z_\phi^y(x + \eta)]$$
$$\geq \mathbb{E}_\eta[z_\theta^y(x + \eta) - z_\phi^y(x + \eta)]$$
$$\geq -\mathbb{E}_\eta[z_\phi^y(x + \eta) - z_\theta^y(x + \eta)]$$

This implies that maximizing $\mathbb{E}_\eta[z_\theta^y(x + \eta)]$ is equivalent to minimizing $\mathbb{E}_\eta[z_\phi^y(x + \eta) - z_\theta^y(x + \eta)]$.

# B    Additional Results

## B.1    Higher Noise Level

In the main paper, we conduct experiments on CIFAR-10 using noise level $\sigma = 0.25$ only. Here, we report our main set of results on CIFAR-10 (Table 3) using higher $\sigma$ values. In Table 8, we report results using $\sigma = 0.5$ and in Table 9, we report results using $\sigma = 1.0$.

## B.2    Using ViT [6]

In the main paper, we used Convolutional Neural Network (CNN) based architectures. However, there is another recently developed class of architectures that use Transformers for the task of image classification [6, 24, 4]. In Table 10, we present results measuring the effectiveness of CRT in transferring robustness from ResNet110 (a CNN-based classifier) to ViT [6] (a Transformer-based classifier). For comparison, we also report results obtained on training ViT with SmoothMix. CRT trained ViT classifiers perform comparable or better than their SmoothMix counterparts.

## B.3    Training SmoothMix Classifiers with CRT Hyperparameters

For generating results using prior methods, we strictly adhere to the hyperparameters reported by them. However, the hyperparameters that we use for CRT training is different than the ones used by prior methods (see Table 16). In Table 11, we report results obtained on using CRT training hyperparameters with SmoothMix. We note that there is not a significant difference in the robustness achieved using the two sets of hyperparameters.

## B.4    Training Teacher with Consistency Regularization [16]

For our main set of results on CIFAR-10, we focused on SmoothMix. In Section 5.4, we discussed Consistency Regularization recently proposed by Jeong & Shin [16] as another method to attain certifiably robust classifiers with at a cost comparable to standard training depending on the setting. Here, we show results when Consistency Regularization is used to train the teacher classifier in Tables 12 and 13. For comparison, in Table 12, we also report the results for training the classifiers using Consistency Regularization. As with other training methods, CRT is effective in transferring the robustness of the teacher classifier irrespective of the training method.

## B.5    ImageNet Results

In Table 14, we present an additional ImageNet result using a different student-teacher pair. We note that, as in Table 7, CRT remains effective.

## B.6    Gaussian Data Augmentation Baseline

Along with the theoretical framework for creating certifiably robust image classifiers using randomized smoothing, Cohen *et al.* [3] also proposed a simple yet effective method for training classifiers with high certified robustness within this framework. This method involves training the base classifier with Gaussian data augmentation. To date, this method remains the fastest way to train classifiers with non-trivial certified robustness using the randomized smoothing framework. However, this method is not as sophisticated as the more recently proposed methods, and so yields much poorer certified robustness than them. Since our work is focused at accelerating certified robustness training, in Table 15 we include the certified robustness of training time results for Gaussian data augmentation baseline for a more thorough comparison.

Table 8: The certified robustness of classifiers with different architectures trained on CIFAR-10 using SmoothMix [15] and CRT. We use CRT to transfer the robustness of a ResNet110 trained using SmoothMix. We report certified test accuracy at different values of $\ell_2$ radius and the Average Certified Radius (ACR). The noise level $\sigma$ is set to $0.5$.

| ARCHITECTURE | 0.00 | 0.25 | 0.50 | 0.75 | 1.00 | 1.25 | 1.50 | 1.75 | ACR |
|---|---|---|---|---|---|---|---|---|---|
| SMOOTHMIX [15] | | | | | | | | | |
| RESNET110 [11] | 65.01 | 57.71 | 49.99 | 42.74 | 35.98 | 29.43 | 23.52 | 17.33 | 0.725 |
| RESNEXT29-2x64D [36] | 63.90 | 56.81 | 48.80 | 40.79 | 33.29 | 27.24 | 20.80 | 14.60 | 0.687 |
| DLA [38] | 65.76 | 58.47 | 51.16 | 43.97 | 37.16 | 30.50 | 23.97 | 18.02 | 0.742 |
| REGNETX_200MF [27] | 64.75 | 57.48 | 49.96 | 42.57 | 35.23 | 28.79 | 22.78 | 16.56 | 0.716 |
| CRT (RESNET110 TEACHER) | | | | | | | | | |
| RESNEXT29-2x64D [36] | 64.89 | 57.81 | 50.63 | 43.39 | 36.49 | 30.07 | 23.92 | 17.40 | 0.732 |
| DLA [38] | 65.23 | 58.33 | 51.23 | 44.04 | 37.09 | 30.47 | 24.39 | 18.37 | 0.743 |
| REGNETX_200MF [27] | 65.35 | 58.18 | 50.87 | 43.74 | 36.83 | 30.33 | 24.17 | 18.04 | 0.739 |

Table 9: The certified robustness of classifiers with different architectures trained on CIFAR-10 using SmoothMix [15] and CRT. We use CRT to transfer the robustness of a ResNet110 trained using SmoothMix. We report certified test accuracy at different values of $\ell_2$ radius and the Average Certified Radius (ACR). The noise level $\sigma$ is set to $1.0$.

| ARCHITECTURE | 0.00 | 0.25 | 0.50 | 0.75 | 1.00 | 1.25 | 1.50 | 1.75 | 2.00 | 2.25 | ACR |
|---|---|---|---|---|---|---|---|---|---|---|---|
| SMOOTHMIX [15] | | | | | | | | | | | |
| RESNET110 [11] | 47.93 | 43.46 | 38.43 | 33.24 | 29.05 | 25.05 | 21.55 | 18.12 | 15.19 | 12.40 | 0.730 |
| RESNEXT29-2x64D [36] | 46.81 | 41.38 | 36.27 | 31.46 | 27.15 | 22.85 | 19.36 | 16.27 | 13.24 | 10.49 | 0.667 |
| DLA [38] | 49.40 | 44.13 | 38.86 | 33.94 | 29.27 | 25.14 | 21.63 | 18.38 | 15.29 | 12.49 | 0.738 |
| REGNETX_200MF [27] | 47.32 | 42.87 | 38.25 | 33.33 | 29.17 | 25.80 | 21.99 | 18.51 | 15.66 | 13.05 | 0.743 |
| CRT (RESNET110 TEACHER) | | | | | | | | | | | |
| RESNEXT29-2x64D [36] | 48.03 | 43.41 | 38.56 | 33.15 | 28.92 | 25.29 | 21.43 | 18.40 | 15.03 | 12.20 | 0.728 |
| DLA [38] | 48.38 | 43.67 | 38.82 | 33.76 | 29.4 | 25.53 | 21.94 | 18.71 | 15.32 | 12.61 | 0.741 |
| REGNETX_200MF [27] | 48.18 | 43.53 | 38.68 | 33.62 | 29.30 | 25.44 | 21.86 | 18.45 | 15.29 | 12.46 | 0.735 |

# C  Training Details

In this section, we provide all these details required to reproduce the results presented in the paper. We begin by reporting the hyperparameters in Appendix C.1 followed by code links and other necessary instructions in Appendix C.2.

## C.1  Hyperparamters

First, we provide details regarding training hyperparameters for our experiments in Table 16. For all training, we perform regularization using a weight decay factor of $1e-4$. Also for all training, learning rate is decayed by a factor of $0.1$ at two pre-determined epochs (see column 'LR Decay' in Table 16). Next, we report method-specific hyperparameters in Table 17. Note that CRT does **NOT** introduce any new hyperparameters.

## C.2  Reproducing Results From This Paper

For reproducing results using SmoothAdv [8], MACER [9], SmoothMix [10], and Consistency [11], we follow instructions provided by the authors and use their respective codes. For CRT, all necessary instructions and code required to reproduce the results are available at https://github.com/Ethos-lab/crt-neurips22.

---

[8] https://github.com/Hadisalman/smoothing-adversarial
[9] https://github.com/RuntianZ/macer
[10] https://github.com/jh-jeong/smoothmix
[11] https://github.com/jh-jeong/smoothing-consistency

Table 10: The certified robustness of a ViT classifier trained on CIFAR-10 using SmoothMix [15] and CRT for different $\sigma$ values (*i.e.,* noise levels). We use CRT to transfer the robustness of a ResNet110 trained using SmoothMix. We report certified test accuracy at different values of $\ell_2$ radius and the Average Certified Radius (ACR).

| $\sigma$ | 0.00 | 0.25 | 0.50 | 0.75 | 1.00 | 1.25 | 1.50 | 1.75 | 2.00 | 2.25 | ACR |
|---|---|---|---|---|---|---|---|---|---|---|---|
| | | | | | SMOOTHMIX [15] | | | | | | |
| 0.25 | 69.38 | 56.65 | 42.34 | 28.47 | 0.00 | 0.00 | 0.00 | 0.00 | 0.00 | 0.00 | 0.415 |
| 0.50 | 50.56 | 43.87 | 37.20 | 30.55 | 24.55 | 19.41 | 14.78 | 10.43 | 0.00 | 0.00 | 0.515 |
| 1.00 | 35.95 | 31.55 | 27.63 | 23.84 | 20.36 | 17.09 | 14.16 | 11.87 | 9.85 | 8.05 | 0.509 |
| | | | | | CRT (RESNET110 TEACHER) | | | | | | |
| 0.25 | 69.63 | 56.60 | 42.29 | 28.45 | 0.00 | 0.00 | 0.00 | 0.00 | 0.00 | 0.00 | 0.415 |
| 0.50 | 60.64 | 53.62 | 46.07 | 39.49 | 32.13 | 25.59 | 19.76 | 14.36 | 0.00 | 0.00 | 0.653 |
| 1.00 | 41.76 | 37.19 | 32.55 | 28.43 | 24.67 | 20.89 | 17.48 | 14.91 | 12.29 | 9.79 | 0.610 |

Table 11: The certified robustness of classifiers with different architectures trained on CIFAR-10 using SmoothMix [15]. Here, we use the same training hyperparameters as the ones we used to train CRT classifiers (see Table 16). We report certified test accuracy at different values of $\ell_2$ radius and the Average Certified Radius (ACR). The architectures are sorted chronologically based on published date. The noise level $\sigma$ is set to 0.25.

| ARCHITECTURE | 0.00 | 0.25 | 0.50 | 0.75 | ACR |
|---|---|---|---|---|---|
| RESNET110 [11] | 78.22 | 69.23 | 58.71 | 46.61 | 0.559 |
| RESNEXT29-2X64D [36] | 77.22 | 66.72 | 55.06 | 42.43 | 0.528 |
| DLA [38] | 78.07 | 69.37 | 58.61 | 46.70 | 0.559 |
| REGNETX_200MF [27] | 77.39 | 68.06 | 57.25 | 45.44 | 0.547 |

Table 12: The certified robustness of classifiers with different architectures trained on CIFAR-10 using Consistency Regularization [16] and CRT. We use CRT to transfer the robustness of a ResNet110 trained using Consistency Regularization. We report certified test accuracy at different values of $\ell_2$ radius and the Average Certified Radius (ACR). The noise level $\sigma$ is set to 0.25.

| ARCHITECTURE | 0.00 | 0.25 | 0.50 | 0.75 | ACR |
|---|---|---|---|---|---|
| | CONSISTENCY REGULARIZATION [16] | | | | |
| RESNET110 [11] | 75.89 | 68.02 | 58.04 | 46.84 | 0.552 |
| RESNEXT29-2X64D [36] | 74.99 | 65.96 | 55.46 | 43.01 | 0.528 |
| DLA [38] | 76.76 | 67.88 | 57.36 | 45.86 | 0.547 |
| REGNETX_200MF [27] | 75.48 | 66.76 | 55.96 | 43.76 | 0.534 |
| VIT [6] | 60.52 | 52.17 | 42.87 | 34.08 | 0.416 |
| | CRT (RESNET110 TEACHER) | | | | |
| RESNEXT29-2X64D [36] | 75.72 | 68.46 | 58.85 | 47.30 | 0.557 |
| DLA [38] | 76.61 | 69.18 | 59.59 | 48.35 | 0.565 |
| REGNETX_200MF [27] | 76.18 | 68.42 | 58.85 | 47.64 | 0.558 |
| VIT [6] | 73.00 | 64.50 | 53.69 | 42.33 | 0.515 |

Table 13: The certified robustness of classifiers with different architectures trained on CIFAR-10 using CRT recursively. The initial classifier was trained using Consistency Regularization [16]. We report certified test accuracy at different values of $\ell_2$ radius and the Average Certified Radius (ACR). Here, the previous generation classifier is used to train the current generation one. Chain length represents the number times CRT was used in training. The noise level $\sigma$ is set to $0.25$. CRT remains effective despite recursive use.

| ARCHITECTURE | CHAIN LENGTH | 0.00 | 0.25 | 0.50 | 0.75 | ACR |
|---|---|---|---|---|---|---|
| RESNEXT29-2X64D [36] | 1 | 75.72 | 68.46 | 58.85 | 47.30 | 0.557 |
| DLA [38] | 2 | 76.36 | 69.20 | 59.35 | 48.46 | 0.565 |
| REGNETX_200MF [27] | 3 | 76.07 | 68.54 | 58.66 | 47.58 | 0.559 |
| VIT [6] | 4 | 74.42 | 66.18 | 55.94 | 44.88 | 0.535 |

Table 14: ImageNet result using CRT and SmoothMix on an additional student-teacher pair. We report the ACR of the ResNet50 teacher and its RegNetX-4.0G student. For reference, we also report robustness of a RegNetX-4.0G network trained independently using SmoothMix. The noise level $\sigma$ is set to $0.5$. CRT remains effective on ImageNet even with a different student-teacher pair.

| TEACHER (RESNET50) | | | STUDENT (REGNETX-4.0G) | |
|---|---|---|---|---|
| TRAINING METHOD | ACR | | TRAINING METHOD | ACR |
| SMOOTHMIX [15] | 0.799 | $\rightarrow$ | CRT | 0.788 |
| STUDENT TRAINED DIRECTLY | | | SMOOTHMIX [15] | 0.877 |

Table 15: Certified robustness and total time of a ResNet110 classifier trained on CIFAR-10 using Gaussian data augmentation. The noise level $\sigma$ is set to $0.25$.

| NETWORK | 0.00 | 0.25 | 0.50 | 0.75 | ACR | TOTAL TIME (H) |
|---|---|---|---|---|---|---|
| RESNET110 | 0.486 | 81.41 | 67.75 | 49.67 | 32.37 | 4.80 |
| RESNEXT29-2X64D | 79.71 | 66.06 | 48.67 | 31.09 | 0.474 | 4.55 |
| DLA | 81.30 | 69.53 | 54.48 | 37.81 | 0.521 | 3.08 |
| REGNETX_200MF | 80.53 | 67.05 | 50.32 | 32.72 | 0.487 | 3.05 |
| VIT | 0.211 | 48.77 | 32.70 | 18.78 | 8.98 | 4.78 |

Table 16: Training hyperparameters used to train classifiers using different methods. For prior works, we closely follow the hyperparameters reported by them. For CRT, we tune hyperparameters to train till convergence.

| METHOD | EPOCHS | BATCH SIZE | INITIAL LR | LR DECAY |
|---|---|---|---|---|
| | CIFAR-10 | | | |
| SMOOTHADV [29] | 150 | 256 | 0.1 | 50, 100 |
| MACER [39] | 440 | 64 | 0.01 | 200, 400 |
| SMOOTHMIX [15] | 150 | 256 | 0.1[*] | 50, 100 |
| GAUSSIAN AUGMENTATION [3] | 200 | 128 | 0.1 | 100, 150 |
| CRT | 200 | 128 | 0.1 | 100, 150 |
| | IMAGENET | | | |
| SMOOTHADV [29] | 90 | 400 | 0.1 | 30,60 |
| MACER [39] | 120 | 256 | 0.1 | 30,60,90 |
| SMOOTHMIX [15] | 90 | 400 | 0.1 | 30,60 |
| CRT | 90 | 400 | 0.1 | 30,60 |

[*] *For SmoothMix training of ViT, we use initial LR of $0.01$ as otherwise training doesn't converge.*

Table 17: Method-specific hyperparameters used in our experiments on CIFAR-10 and ImageNet.

| $\sigma$ | METHOD | HYPERPARAMETER DETAILS |
|---|---|---|
| | | CIFAR-10 |
| 0.25 | SMOOTHADV [29] | 8-SAMPLES, 10-STEP PGD ATTACK WITH $\epsilon = 1.0$ |
| | MACER [39] | $k = 16, \lambda = 12.0, \beta = 16.0, \gamma = 8.0$ |
| | SMOOTHMIX [15] | $T = 4, m = 2, \eta = 5.0, \alpha = 0.5$ |
| | CONSISTENCY [16] | $\lambda = 20, m = 2, \eta = 0.5$ |
| 0.5 | SMOOTHMIX [15] | $T = 4, m = 2, \eta = 5.0, \alpha = 1.0$ |
| 1.0 | SMOOTHMIX [15] | $T = 4, m = 2, \eta = 5.0, \alpha = 2.0$ |
| | | IMAGENET |
| 0.5 | SMOOTHADV [29] | 1-SAMPLE, 1-STEP PGD ATTACK WITH $\epsilon = 1.0$ |
| | MACER [39] | $k = 2, \lambda = 3.0, \beta = 16.0, \gamma = 8.0$ |
| | SMOOTHMIX [15] | $T = 1, m = 1, \eta = 1.0, \alpha = 8.0$ |