# OpenReview forum: "Accelerating Certified Robustness Training via Knowledge Transfer"
_NeurIPS.cc/2022/Conference — NeurIPS 2022 Accept_

### Official Review · Reviewer_iNdk · 2022-06-28

**Rating:** 5
**Confidence:** 4
**Soundness:** 4 excellent
**Presentation:** 4 excellent
**Contribution:** 2 fair

**Summary:**

This paper proposes certified Robustness Transfer (CRT) for reducing the computational overhead of any certifiably robust training method through knowledge transfer. The paper targets a scenario where different neural network architectures evolve, and one needs to retrain different architectures with time-consuming, certifiably robust training methods.

**Questions:**

1. In the recursive evaluation of CRT in table 3, the result of chain length 3 became worse than shorter chains. Does it imply that the effectiveness of CRT will decay with respect to the number of recursions? If it does decay, then it will harm the speed up brought by CRT because one needs to retrain after applying CRT two or three times.
2. Why not directly minimizing $z^{y_i}_\phi(x_i + \eta) - z^{y_i}_\theta(x_i + \eta)$ but minimizing the distance between them?
3. Algorithm 1 only minimizes $\|z^{y_i}_\phi(x_i + \eta) - z^{y_i}_\theta(x_i + \eta)\|_2$. How about incorporating other losses?

**Limitations:**

Yes, the authors adequately addressed the limitations in section 5.5.

**Strengths And Weaknesses:**

Strength:
+ The paper conducts solid experiments to demonstrate the effectiveness and efficiency of the CRT framework.

Weakness:
- The paper lacks novelty. The approach of using transfer learning to improve (certified) accuracy and training efficiency is not new. The fact that transfer learning can transfer the robustness is already explored by existing works, as mentioned in section 3.3. The paper provides a theatrical analysis of the transferability, but the analysis is trivial.
- Another concern is that it is unclear whether the robust knowledge has been transferred to the student model apart from metrics like certified accuracy and ACR. In other words, the paper would be improved by theoretically analyzing the decay of the robustness transfer, e.g., after $r$ recursions, specific robust metrics would decay at rate $h(r)$.


Detailed comments:
1. For all tables presenting certified metrics and speeds, please show the baseline training method with data augmentation by Gaussian noise. Comparing this baseline method can help illustrate the necessity of fancy training methods and CRT.
2. The paper does not explore how far transfer learning can go without considering the cost of training, e.g., is it possible to improve algorithm 1 such that it incorporates the existing loss (capturing the knowledge of the teacher) and another loss that uses fancy certifiably robust training methods or the baseline Gaussian noise data augmentation.


===============================

Post one round of rebuttal:

Changed Contribution from 1 to 2

Changed Rating from 3 to 4

===============================

After reading other reviewer and authors discussion about experiments

Changed Rating from 4 to 5

---

> ### Author Response · Authors · 2022-08-01
> **Response for Reviewer iNdk**
>
> Thank you for your valuable feedback. Following is our response to your questions and concerns:
>
> - **Lack of Novelty/Transferrable Robustness is not New.**
>
> As we discussed in Section 3.3 the paper, prior works have explored the transfer of adversarial robustness between neural network architectures. However, as we noted, these works limit their study to transferring **empirical robustness**. Our work is novel as we explored how **certifiable robustness** can be transferred between networks. A network which is empirically robust only serves to prove that it is robust to the attack it was evaluated against, i.e. the attack was unable to find an adversarial perturbation. Empirical robustness offers no guarantee that no adversarial perturbation exists for a given input as evidenced by the large body of work on “adaptive adversarial attacks” [1,32]. Certifiable robustness, however, guarantees that the prediction for a given input is consistent within a pre-defined radius irrespective of any adversarial attack. As these two types of robustness are **fundamentally different**, it is not obvious if the observations/findings related to empirical defenses also apply to certified defenses. Our work is the first work to explore and propose a method to inexpensively transfer certifiable robustness between classifier architectures.
>
> - **Add Gaussian Augmentation as an additional baseline comparison.**
>
> We have measured the accuracy and training time of classifiers trained using Gaussian Augmentation, but as it is a simple data augmentation approach, it adds very little overhead compared to standard training. It also results in lower certifiable robustness compared to the works we analyzed in our paper. Following works like SmoothMix, Macer, and SmoothAdv propose modified training objectives and pipeline in order to improve certified robustness achieved. We did not include a comparison to gaussian augmentation as our intention was to develop a method to reduce the computation overhead, while preserving the robustness provided by state-of-the-art certifiable robustness techniques. If necessary, we can re-include the gaussian augmentation results.
>
> - **…the results of chain length 3 became worse than shorter chains. Does it imply that the effectiveness of CRT will decay with respect to the number of recursions?**
>
> This observation is inaccurate. We note that (i) the ACR actually increases between the original teacher and chain length 2, and only slightly decreases in chain length 3, and (ii) all three classifiers presented in Table 3 show higher ACR than the original teacher (RESNET110,  0.550).  Furthermore, if we compare the certified robustness of the classifiers in Table 3 to their respective baseline classifiers trained from scratch in Table 2, we see that even with recursion, CRT trained classifiers are similar or more robust.
>
> However, let us imagine a case where the effectiveness of CRT decays as the number of recursions increases and after **k recursions**, the next classifier generation needs to be trained again from scratch using an expensive certified training method. Even in this scenario, however, CRT would reduce the overhead of state-of-the-art certifiable training methods by approximately a factor of k.
>
> - **Is it possible to improve algorithm 1 such that it combines the existing transfer loss and with a loss from another state-of-the-art certifiable training method?**
>
> It may be possible to improve CRT and its training methodology such that it can utilize a more complex loss objective without compromising its efficiency, however,existing state-of-the-art training methods impose a high training overhead as a result of using more complex training objectives so incorporating the same training objective into CRT while preserving CRT’s training cost is a non-trivial task. Furthermore, CRT currently remains to be the first work that enables efficient transfer of certifiable robustness between neural network architectures so we leave such improvements to future work.
>
> - **Why did you not directly minimize the softmax outputs and instead used an l2 distance loss? More generally, could other loss functions be used in place of the l2 loss?**
>
> Based on equation 6, any loss function that minimizes the difference between the teacher’s output and the student’s output should enable robustness transfer. In our initial investigation, we explored several implementations of CRT using different loss functions. In addition to the l2 loss, we measured certifiable robustness of networks trained using the KL divergence loss, the cosine loss, and the cross entropy loss with the teacher's soft outputs. The l2 loss function resulted in the best certifiable robustness overall. Due to space constraints, we did not include these results, but can provide them as an appendix in the final submission.

---

> > ### Comment · Reviewer_iNdk · 2022-08-02
> > **Author-reviewer Discussion**
> >
> > Thanks for the response. The response well addressed my questions, and I provided the following discussion with the authors.
> >
> > > Lack of Novelty/Transferrable Robustness is not New.
> > After reading the response and the related work in detail, I find my previous judgment inappropriate. I will raise my review score regards this point, and the novelty issue will be neither a strength nor a weakness, in my opinion. I want to elaborate my thoughts more on this point in the discussion process:
> >
> > 1. *it is not obvious if the observations/findings related to empirical defenses also apply to certified defenses.* As stated in the response. I agree that the community needs a paper to make the observation/finding, but such a paper would not interest me if the paper provided no other findings. And such concern is the reason for the second bullet point of the weakness, as I expected more analysis.
> >
> >
> > 2. The certified robustness and the empirical robustness are two measurements of robustness, while the certified one measures the lower bound and the empirical one measures the upper bound of the robust accuracy. The novelty of certified robustness (especially the certified robustness by randomized smoothing in the *original* paper, Cohen et al.) lies in the certification of the smoothed classifier using hypothesis testing or the Neyman-Pearson Lemma. The baseline used in training, i.e., adding Gaussian noise, shares the same idea of data augmentation and adversarial training. Existing works have shown that empirical robustness can be transferred. This paper shows that certified robustness can also be transferred. Combining these two findings might lead to a wilder claim about the transferability of the robustness, but there is no more evidence other than the lower and the upper bound of the robust accuracy (the empirical and certified one). This direction, seemingly deviated from the paper's motivation, might be interesting for future work.
> >
> > > Add Gaussian Augmentation as an additional baseline comparison.
> >
> > I regard the Gaussian baseline as a necessary comparison because they have a similar training time. Although it is a simple baseline (as SmoothMix improves almost 2X of the certified accuracy on CIFAR-10 when r=0.75 as stated in their paper), it is still important for this paper to include it as a sanity check because readers might not know of the detailed improvement of SmoothMix compared to the Gaussian-noise baseline.

---

> > > ### Author Response · Authors · 2022-08-03
> > > **Author-reviewer Discussion**
> > >
> > > We thank the reviewer for engaging in discussion with us. Please find our response to your previous comment below.
> > > 1. In addition to showing that certified robustness is transferable and providing a theoretical justification for the transfer process, our findings also include showing for the first time that: (a) recursive (certified) robustness transfer is possible, and (b) a smaller network can be used to accelerate robustness training of a larger network (Section 5.1). Furthermore, as our paper is the first to study the transferability of certifiable robustness in an effort to reduce training overhead, we focused our analysis on factors affecting the overall computational cost of CRT (e.g. training time reduction, teacher training method, dataset scalability). We agree that a theoretical analysis of the decay would be an interesting result, but, our empirical results in Table 3 did not indicate a clear decay trend, if one exists at all. As noted previously, the performance in Table 3 shows improvements over the SmoothMix baseline.
> > > 2. Indeed, both prior works studying the transferability of empirical robustness and our work studying the transferability of certifiable robustness motivate further study in the general transferability of robustness between models. In fact, we believe a broader study of transferability across multiple AI metrics (e.g. fairness, robustness, explainability, privacy, etc) could be done. Such a study would be of great use given the rising interest in foundational models, which rely heavily on transfer learning.
> > >
> > > We will re-include the Gaussian Augmentation results as an additional baseline to the paper so readers have context around robustness improvements made by prior work like Smoothmix. As an example of the results we can include, the ACRs of the ResNet110, DLA, and RegNetX_200MF networks trained using GDA are 0.461, 0.520, and 0.487 respectively.

---

### Official Review · Reviewer_aqwd · 2022-07-09

**Rating:** 5
**Confidence:** 3
**Soundness:** 2 fair
**Presentation:** 2 fair
**Contribution:** 2 fair

**Summary:**

I think the paper is well-motivated and the proposed method is straightforward. This topic is interesting. However, the empirical contribution is over-claimed to me by a wrong measurement and in fact they might be not significant and lack enough experiments. I therefore vote for rejection for now but am open to changing depending on the authors’ response.

**Questions:**


My questions are embedded in my comments about the weakness. I can summarize them again in this part but please find them in the context above.

1) Why is this work significant as students are no significantly better than teachers and the robustness transfer is only between architectures?

2) Why is the training cost of CRT only the cost of training students, but is used to compare with baselines that actually train teachers ?


**Limitations:**

The paper has a separate paragraph to discuss its limitations.

**Strengths And Weaknesses:**

### Strength
The writing of this paper is pretty clean and easy to follow. The motivation is very clear and the presentation of the proposed method carries that clearness.

### Weakness
My major concern for this paper is that the empirical contribution is over-claimed. However, Section 5.1 is the place I think the authors measure their work in a correct way but the corresponding results are neither significantly better nor comprehensive enough to support their claimed contribution. I will elaborate.

**Training time comparisons are unfair.**  The cost of the CRT pipeline in terms of the per-epoch time is measured in the wrong way. The current way, excluding Section 5.1, to measure the training cost for CRT is total cost = student cost. If this paper is about to compare the training cost between baselines that train student networks, this is correct. However, all the baselines are methods that actually train the teacher network. Therefore, the correct cost of CRT should be *total cost = teacher cost + student cost*. It is okay to assume a robust teacher network exists but not to assume that the cost of having a teacher network is zero. The authors seem to have noticed this problem. I found the correct measurements in Section 5.1. In fact, if this paper follows the way Section 5.1 is designed, the results will be very impressive: only do robust training on a tiny/small network and use them as teachers for monster networks, which can save a significant amount of time. To do this, the current experiments in Section 5.1 is far from enough. As one may notice that the training cost of the teacher dominates the training budget of CRT because training students do not need robustness regularizations. If one ignores the training cost of teachers, this paper is just to compare the training cost between robustness regularization and standard training, which is not interesting.

**The significance of the work needs more justifications.** All the results show that the student networks are just marginally better or worse than the teachers. This raises a question: why in practice does one want to do this robustness transfer? Why do we not directly use the teacher network? The transfer is only between architectures and not between datasets. Unless network architectures are limited in some sense or I missed something, I don’t see why we need CRT to produce a similar network.

**The discussion on the scalability of CRT is confusing.** Section 5.3 aims to show the scalability of CRT, which is pretty confusing to me. My superficial understanding of the scalability of CRT is determined by the scalability of training teachers and the scalability of training students. The scalability of teachers is determined by the robustness training method proposed by other work. The training of student models is just standard training, which always scales up. However, if the authors take my advice to shift the focus of this paper to train large robust student networks from small teachers, then it is fair to claim that CRT is the way to scale robustness training for large networks.

**Some figures and tables are not necessary.** I find Figure 1 and Table 1 are not necessary at all because 1) I don’t see any information related to the topic this paper tries to discuss in Figure 1. This is just a very general plot for deep learning, and 2) making the factors in a table does not help convey more messages than pure text. There is no more information at all.

---

> ### Author Response · Authors · 2022-08-01
> **Response for Reviewer aqwd**
>
> Thank you for your valuable feedback. Following is our response to your questions and concerns:
>
> - **Motivation behind using CRT**
>
> Figure 1 highlights the progression of neural network architectures over time. As it demonstrates, newer architectures result in either improving performance or reducing network size compared to prior generations. While previous works have explored transfer of empirical robustness; prior to our work, **certifiable robustness** was not possible to be preserved between architecture generations. Therefore, in order to provide certifiable guarantees of  the security and safety of deployed machine learning applications between architecture generations, **it was necessary to re-train the new architectures from scratch**. CRT enables inexpensive training of certifiably robust new generation classifiers (like ResNeXt) by transferring the certifiable robustness of the current generation classifier (like ResNet), bypassing the need to re-train the new generation using expensive state-of-the-art methods. Our results show that CRT trained classifiers exhibit comparable certified robustness to classifiers trained with state-of-the-art methods, at a fraction of their training time, even when CRT is used recursively.
>
> - **Regarding training time comparison**
>
> As noted by the reviewer, the first set of CRT training time results in Table 4 does not include the time required to train the ResNet110 teacher. Recall that we designed CRT to enable the transfer of certifiable robustness between networks. Therefore, an expensive SOTA certifiable training method **only needs to be performed once**. After which, for all successive architecture generations, CRT can be used to inexpensively train a certifiably robust classifier. The cost of training the original ResNet110 teacher, therefore, gets **amortized**. As an example, if we were to compare the computation time required to train all four classifiers in Table 4 (including the ResNet110 classifier), Smoothmix requires 18.98+45.21+35.6+15.39 = **115.98 hours** and CRT requires 18.98+4.8+3.46+3.44 = **30.68 hrs**, representing a time savings of **~73.5%**. We note that even when we add the time required to train the ResNet110 teacher to each CRT classifier in Table 3, CRT is faster in all, but one case. While we recognize that CRT can also be used to train large robust classifiers by first training a smaller robust classifier, this is a side benefit of our main contribution, which is the ability to transfer certified robustness between generations of neural network architecture.
>
> - **Regarding scalability of CRT**
>
> In Section 5.3, we use scalability to refer to the complexity of the input, dataset, and classification task. Prior adversarial robustness works, especially defenses, often only demonstrated successful results on smaller datasets like MNIST and CIFAR10. A reproducibility issue sometimes arose when deployed on a larger dataset, often ImageNet [a]. We include Section 5.3 to show that CRT remains effective, even on larger datasets.
>
> **References**
>
> [a] Kurakin et al., “Adversarial Machine Learning At Scale”, ICLR, 2017.

---

> > ### Comment · Reviewer_aqwd · 2022-08-03
> > **Follow-ups**
> >
> > Thank you for your response to my initial review.
> >
> > After reading the response and reading feedback from other reviewers, I still have a few questions. Please find them below.
> >
> > **Motivations.** Reviewer t68V actually has a better summary for the contribution of this submission than me: this is a work to do transfer learning for robust classifiers. Reviewer t68V highlights the limitation in the novelty of the paper while acknowledging the practical impact of the work. In the response, the authors mention that we want a new architecture, e.g. ResNeXt, to be as robust as an old architecture, e.g. ResNet. Therefore, my follow-up question is, can the author provide some use case such that, given a robust ResNet, an untrained ResNeXt, a dataset and a task, CRT can produce a robust ResNeXt on the same task and dataset. However, the robustness of ResNext may just match the robustness of ResNet or even worse. Under this circumstance, what is the use case or motivation that one may still want to use ResNeXt?
> >
> > **Timing.** I am not sure whether I fully understand how the numbers are calculated. Can the author provide a tiny table (can be just one row) to compare things column by column? Besides, I still think the paper is trying to compare the cost between transfer learning, e.g.. CRT, with robust training, e.g. Smoothmix. After all, transfer learning is solving a different problem as learning robust classifiers from scratch. This may be because there are no other baseline work that does robustness transfer learning so the paper has to compare itself with Smoothmix. With that being said, showing transfer learning is faster by using same architectures is less interesting to m because this is obvious.
> >
> > **Scalability.** I am not sure I understand the authors' response regarding my comments on the scalability. I think my question is only about the authors' claim that CRT scales up because it works for bigger architectures with affordable memory and time (which is obvious because this is transfer learning). I am not sure how reproducibility and dataset complexity are related? Can the authors elaborate ?
> >
> > -- The following comment is not a question or something needs addressing. --
> >
> > **"Side-effect".** Though the authors believe transferring robustness from small architectures to larger ones is a side product, my personal view is that this is not trivial because this would allow users with limited resources to also have access to bigger robust models. For example, use a small CNN to teach vision transformer models in the future.

---

> > > ### Author Response · Authors · 2022-08-04
> > > **Response for Reviewer aqwd**
> > >
> > > We thank the reviewer for engaging in discussion with us. Please find further clarifications based on the last comment below.
> > >
> > > ### Motivations
> > > Developers are motivated to upgrade their models because newer models provide higher accuracy or preserve accuracy while reducing network size (as seen in Figure 1). In safety and privacy critical applications, developers are also motivated to preserve adversarial robustness across generations of models as cheaply as possible. While prior solutions require re-training newer models from scratch, CRT enables re-use of older robust models to accelerate training of newer **robust** models.
> > >
> > > **“What happens if robustness is equal to or worse …”**
> > >
> > > We think the confusion of the reviewer is likely due to the fact that the results in the paper (Tables 2 and 3) doesn’t always show a positive trend in certified robustness or clean accuracy across newer generations of models when using Smoothmix or CRT. For example, in Table 2, training a ResNeXt classifier using SmoothMix yields lower ACR (0.516) than training a ResNet110 classifier using SmoothMix (0.550), even though ResNext is the newer generation model. There is another case of this in Table 3, where the RegNetX model’s robustness is worse than the DLA model. We note that the main focus of our paper was **if transfer of certified robustness is possible**. Wherever possible, we use the training hyperparameters that are consistently used by prior works [3,29,39,15], in order to provide a fair comparison and aid reproducibility of our results. The results in our paper show, for the first time, that **certified robustness transfer is possible**, even without the need for hyperparameter tuning. Having said that, we expect that with proper hyperparameter tuning, the performance of all trained networks (for both SmoothMix and CRT) would show a positive trend as in Figure 1. In other words, a developer switching to ResNext from ResNet would observe improved clean accuracy and certified robustness (cheaply, when using CRT).
> > >
> > > ### Timing
> > > To clarify, in the paper we use the phrase “knowledge transfer” as CRT transfers the robust information learned by the teacher to the student. However, CRT does so using the **knowledge distillation framework**. Unlike in transfer learning, where the reduced training time can be attributed to pre-training on a large source set, that in turn enables fine-tuning on a smaller dataset for fewer training epochs, CRT reduces training overhead by replacing SOTA certifiable robust training with standard training, while preserving SOTA certified robustness. It is not immediately obvious whether knowledge distillation will transfer certain desirable properties, specifically certifiable robustness of the teacher classifier. Even though empirical robustness has been shown to transfer, these findings don’t trivially translate to certified robustness. This is because works on **empirical and certified robustness are fundamentally different**. Furthermore, knowledge distillation has, to the best of our knowledge, never been used with the goal of accelerating training. Traditionally, a large network is distilled into a smaller one to improve its performance. Our work is the first to show that certifiable robustness can be transferred from a robust teacher, thereby **enabling faster training of robust models** with CRT. Given this, the only baseline we can compare against is training from scratch using SOTA methods.
> > >
> > > Could the reviewer please clarify what information they need in the table they ask for, that is not already available in the main paper (Tables 4 and 5) and our previous response? We'd be happy to provide this information.
> > >
> > > ### Scalability
> > > We believe there is a confusion with regards to what ‘scalability’ means. Having already established that CRT reduces computational cost, in Section 5.3 we test whether **CRT remains effective as we scale the complexity of the dataset**. Such 'scalability' testing is necessary as prior work has found some adversarial defenses break down when evaluated on more complex datasets [a]. In their words:
> > >
> > > “Future defenses need to be evaluated on multiple data sets […] While we have found CIFAR to be a reasonable task for evaluating security, in the future as defenses improve it may become necessary to evaluate on harder datasets (such as ImageNet)”.
> > >
> > > Therefore, as in recent prior works on adversarial defense, we follow this recommendation and include Imagenet results in Section 5.3.
> > >
> > > ### References
> > > [a] Carlini and Wagner, “Adversarial Examples Are Not Easily Detected: Bypassing Ten Detection Methods”, AISec, 2017.

---

> > > > ### Comment · Reviewer_aqwd · 2022-08-09
> > > > **Will increase the score**
> > > >
> > > > Thanks for addressing my concerns and I think I better understand the paper's contributions. I will increase my score from 3 to 4 as I have mis-understood the the "scalability" of the work. I think the empirical contributions of the work may be useful with more tuning. The presentation in the current paper shows the certified robustness can be transferred as well. Therefore, I decide to increase my score.

---

### Official Review · Reviewer_w1QE · 2022-07-11

**Rating:** 7
**Confidence:** 4
**Soundness:** 3 good
**Presentation:** 3 good
**Contribution:** 3 good

**Summary:**

This paper presents a method for more efficient re-training of certifiably robust models by means of knowledge distillation. Specifically, the proposed method minimizes the l2 distance between the student and teacher softmax predictions on Gaussian-noise perturbed inputs, where the teacher is a certifiably robust model trained.

**Questions:**

- How is clean accuracy affected?

Suggestions:
- The motivation of the paper could be discussed and evaluated in more detail. For example, the scenario considered in this paper is the need to re-deploy a different architecture using the same dataset, but it would also be interesting to explore fine-tuning the student network on new data.
- It would still be helpful to include in Tables 6 and 7 the certified robustness of the student architecture when trained from scratch in order to better calibrate the results.
- The consistency regularization method mentioned in section 5.4 should also be referenced in the background section at the beginning of the paper.


**Limitations:**

Yes

**Strengths And Weaknesses:**

Strengths
- The paper empirically shows that the proposed method significantly speeds up training a certifiably robust model on CIFAR-10 (with the constraint of requiring access to a certifiably robust teacher model), and can scale to ImageNet.
- The contributions and motivation of this paper are made clear.
- The writing is straightforward.
- The distilled models have better certified robustness than those trained from scratch given the same network architecture, even when the model is a “second-generation” distillation.
- The transfer of certified robustness using distillation is a novel application of distillation.

Weaknesses
- The related work is lacking in terms of prior works on knowledge distillation - it would be useful for the reader to include a section on knowledge distillation and relate the technique used in this paper to existing works.
- The experimental evaluation could be somewhat more rigorous, and the proposed method explicitly compared to the consistency regularization method referenced in section 5.4.

---

> ### Author Response · Authors · 2022-08-01
> **Response for Reviewer w1QE**
>
> Thank you for your valuable feedback. The clean accuracy of our classifiers, as we define on line 199, is given by the certified accuracy at $r=0.0$. Our expression of clean accuracy in this way is adopted from the foundational work by Cohen et al [3], but we will strive to clarify this in the final submission. Based on these results, classifiers trained using our approach have similar clean accuracy as those trained from scratch using state-of-the-art certified robust training methods. With respect to expanding our discussion of prior works to include knowledge distillation and consistency regularization, we have omitted the discussion of these topics due to space constraints but will add them to the background section as requested in the final version of the paper. Finally, we will include an additional row in Tables 6 and 7 to show the performance of the student architecture when trained from scratch.

---

### Official Review · Reviewer_t68V · 2022-07-17

**Rating:** 6
**Confidence:** 5
**Soundness:** 4 excellent
**Presentation:** 4 excellent
**Contribution:** 3 good

**Summary:**

In this work, the author study the transfer learning setting for training a robust classifier using randomized smoothing. It relies on a pretrained smooth model, such as MACER and SmoothAdv, as the teacher model, and proposes a simple loss function to make the student model also learn a similar smoothness. With a good teacher model, the student model can be trained very quickly, much faster than training from scratch. Adequate experiments on CIFAR-10 and ImageNet are conducted with ablation studies to show the effectiveness of this approach, where the improvement is mostly on training time. Even in the setting where a teacher model is not available, the authors show that the training a smaller teacher model first can still be beneficial.

**Questions:**

Although this paper proposes a quite simple algorithm but I feel it is solving a real problem and the proposed approach is effective. Thus, I tend to accept this paper. I have no major questions regarding this paper.


**Limitations:**

Limitations are well discussed and I like the discussions on page 9.

**Strengths And Weaknesses:**

Strength:

1. Thorough empirical results with good speed up on training time, even on large datasets (ImageNet) and large models (ViT-B).
2. Method is simple, effective, and sound.
3. The paper is well presented and easy to understand.

Weakness:

The novelty of this work is limited, because it simply combines transfer learning with robust training using a simple loss function, without further theoretical justifications. However, I feel it is solving a real problem that training a certifiably robust model can be slow, and using a teacher model can indeed speedup the training of large robust model by a large factor without sacrificing too much performance.

---

### Meta-Review · Area_Chair_cFD4 · 2022-08-22

**Recommendation:** Accept
**Confidence:** Less certain

**Metareview:**

This work considers how to transfer a well-trained, certified robust model (i.e., randomized smoothing model) with data from a new domain. To achieve this, it uses a pre-trained smooth model as the teacher model and proposes a simple loss function to make the student model also learn a similar smoothness. With a good teacher model, the student model can be trained very quickly, much faster than training from scratch.

Although some reviewers and I found the novelty of the work is limited, all of us agree that this work has empirical values in how to efficiently train a robust neural network in practical scenarios. Therefore, I recommend acceptance.

**Award:**

No

---

### Decision · Program_Chairs · 2022-09-14

Accept